# The Experience of Spanish Prison Nurses with the HIV Pandemic

**DOI:** 10.3390/healthcare12020184

**Published:** 2024-01-12

**Authors:** Enrique J. Vera-Remartínez, Jose Antonio Zafra-Agea, Julio Garcia-Guerrero, Maria Pilar Molés-Julio

**Affiliations:** 1Penitentiary Center of Castellón I, 12006 Castellón, Spain; enriquevera@gmx.es; 2Department of Nursing, Faculty of Health Sciences, UManresa, Fundació Universitària del Bages, Universitat de Vic, Universitat Central de Catalunya, 08242 Manresa, Spain; 3The Bioethics Committee of the Castellón Health Department, E-12071 Castellón, Spain; garciaj@comcas.es; 4Department of Nursing, Faculty of Health Sciences, Universitat Jaume I, 12071 Castellón, Spain; mjulio@uji.es

**Keywords:** acquired immunodeficiency syndrome, nursing, prisons, Spain

## Abstract

Introduction: This study discusses the experiences of nurses in Spanish prisons during the HIV/AIDS pandemic, emphasizing administrative changes and adaptive measures, such as the creation of the Subdirectorate General of Penitentiary Health. Objective: We describe the experiences of nurses in Spanish penitentiary centers in the face of the HIV/AIDS. Materials and methods: The interpretative and phenomenological approach explored experiences between 1981 and 2023 through focus groups and convenience sampling. Participants were recruited through telephone calls and telematic meetings using Microsoft Teams. Four key themes were identified: the stigmatization of inmates, changes in nursing, the importance of training and resources, and equal treatment between genders. Results: The nursing vision focused on gathering information, providing comprehensive patient support, and sharing personal experiences. Discussion: This research provides a historical perspective on the adaptation of prison nursing to the pandemic, highlighting coping processes and suggesting future lines of research on the experiences of inmates, prison guards, and surveillance officers. Conclusions: We highlight the low media visibility of the pandemic in prisons, underlining the importance of understanding and valuing the nursing experience in prison settings during health crises.

## 1. Introduction

This research aims to address the following question: “What have been the experiences of nurses in Spanish prisons regarding the HIV/AIDS pandemic?”.

Understanding how the HIV/AIDS pandemic influenced administrative changes, staff increases, the provision of material resources, staff training, and even architectural changes in prisons to adapt to the pandemic, with a high-risk population for HIV infection, is essential knowledge that forms part of the history of Spanish prison nursing. This knowledge may prove useful in facing future situations.

Individuals incarcerated in penitentiary institutions represent a marginalized population due to their numbers and social standing. At the onset of the first cases of HIV infection in Spanish prisons, amidst the scientific uncertainty surrounding this condition, significant alarm was raised, instilling fear among both inmates and the custodial staff, leading to the stigmatization of those affected [1].

These were tumultuous times, especially as the first cases spread within prison facilities, resulting in protests initiated by inmates, including some riots, advocating for proper hygienic and sanitary conditions, especially for those infected [2,3,4].

This period also saw formal protests by prison staff, including demonstrations demanding resources to prevent contagion [5,6]. There were even proposals to segregate infected individuals in specific facilities, reminiscent of ‘leper colonies’ [7].

In addition to this, healthcare in Spanish prisons was more than precarious during this time. Obsolete facilities, minimal healthcare presence (not more than three hours of doctor and nurse attendance per day), scarce or nonexistent resources, a lack of medication, and low salaries contributed to a demotivated healthcare staff [8].

The emergence and spread of HIV infection and its correlate, acquired immunodeficiency syndrome (AIDS), within prisons marked a turning point necessitating crucial changes in the penitentiary context.

One significant milestone in these changes was the establishment of the Subdirectorate General of Penitentiary Health (SGSP) in late 1989, at that time under the Ministry of Justice, responsible for establishing an exclusive prison healthcare system parallel to the National Health System (NHS) [8]. The SGSP created its own epidemiological surveillance system and promoted prevention and health education programs for both inmates and staff [8].

Simultaneously, there was an adjustment in schedules and healthcare staff numbers, with a substantial recruitment of doctors and nurses in prison services during certain years. A comprehensive modernization of prison infrastructures was also initiated, replacing outdated facilities that were incapable of fulfilling penitentiary objectives. The Spanish Government approved the Plan for the Amortization and Creation of Penitentiary Establishments (PACEP) in 1991, envisioning the construction of “standard centres” and periodic updates of all prison infrastructures across the country.

Professional training, health education, and both individual and collective talks laid the foundation for prevention programs, especially at a time when no other tools were available to combat the virus. Agreements for specialized healthcare assistance with the NHS were established. Penitentiary patients were visited by specialists, and the first pharmacological treatments began in 1987 with Zidovudine (Retrovir^®^) [9]. Subsequently, new drugs or highly active antiretroviral therapies (HAART) and their combinations emerged, transforming a once deadly disease into a chronic one [9]. Prevention campaigns raised awareness among the incarcerated population, discouraging risky practices, leading to a decrease in prevalence.

While experiences, perceptions, and needs of nurses in specific hospital services have been studied [10], as well as nursing perspectives in other pandemics like the recent COVID-19 pandemic [11], there is limited or no literature on the HIV/AIDS pandemic in the unique setting of prisons. The Spanish print media, reflecting events throughout the different periods of this history, serves as another source referenced in this study to provide context.

This research aims to contribute previously unknown information and serve as a reference for future investigations in the specific field of Spanish prison nursing, particularly during one of the most significant pandemics we have faced. Understanding the past is crucial for facing the future in public health.

Therefore, the primary objective of this research is to describe the experiences of nursing staff in Spanish penitentiary centers during the HIV/AIDS pandemic. It seeks to shed light on how nursing had to adapt to an overwhelming situation in an environment unfamiliar to society.

## 2. Materials and Methods

The design of this research was undertaken from an interpretative paradigm with a phenomenological perspective, focusing on interpreting the experiences of nursing professionals in Spanish penitentiary institutions [12].

**Sample Description:** In qualitative research, the power of the sample does not depend on its size but rather on the extent to which sampling units provide valuable information. The goal is typological representation, not statistical representation [13]. Participants were intentionally and thoughtfully selected to ensure a representative range of opinions and experiences among nursing professionals in penitentiary settings from 1981 to 2023 across various provinces in Spanish autonomous communities.

Convenience sampling based on maximum variation was employed, intentionally selecting a heterogeneous sample to observe commonalities in experiences and identify regularities and peculiarities among professionals. Different groups were formed based on age and years of experience, considering potential differences in perspectives and discourses among professionals.

Nurses from Spanish penitentiary institutions of both genders were included, working in small, medium, and large penitentiary centers. Efforts were made to include informants from emblematic centers such as the now-closed Penitentiary General Hospital of Carabanchel in Madrid (where initial AIDS patients were referred) and the former Model Prison of Barcelona (a reference center at the beginning of the pandemic) to enrich historical insights. Informants worked or had worked in provinces including Barcelona, Castellón, Cuenca, Ibiza, Madrid, Murcia, Seville, Soria, Teruel, and Valencia.

**Selection Procedure:** To recruit participants, initial phone calls were made, obtaining phone numbers mainly from personal phone directories, and in three cases, through acquaintances who contacted the informants. All participants authorized the sharing of their contact information, and no incentives were offered to participants for involvement. Due to geographical dispersion, virtual meetings via Microsoft Teams were scheduled, facilitated by email invitations, and at a convenient time for all participants. In the email, each participant was assigned an anonymous code for confidentiality, and socio-demographic information was requested (age, overall professional experience, experience in prisons, marital status, and current employment status). A document providing information about the research and requesting authorized consent was also provided.

**Data Collection Techniques:** Two discussion groups were established, one comprising professionals who worked at the beginning of the pandemic when effective treatments were unavailable, and the other with professionals who began working when highly active antiretroviral treatments were available. Both groups were homogeneous internally and heterogeneous between them.

Discussion groups consisted of a maximum of six participants, adhering to recommendations suggesting groups of six to twelve participants [14]. A saturation of information was reached with the initial selection, eliminating the need for additional informants.

The moderator guided the discussion based on a pre-established script equivalent to a semi-structured interview, pilot-tested with six professionals to ensure clarity (Table 1). The moderator played a less prominent role, allowing the group to construct the narrative discursively.

All sessions were recorded via video on the Microsoft Teams platform, providing not only audio files but also valuable non-verbal cues from participants. Transcriptions in text format were obtained from the recordings and used as support for analysis through ATLAS.ti v.9, a computer-assisted qualitative data analysis software (CAQDAS).

Quantitative discrete variables included age, overall professional experience, and experience in penitentiaries, expressed in years. Qualitative categorical variables included gender, marital status, and current employment status (active or retired), as shown in Table 2.

**Content Analysis:** Textual corpus preparation involved creating a literal transcription of all content, primarily focusing on information essence (eliminating noise, pauses, non-standard accents, etc.) [15]. The transcript was meticulously reviewed with audio and video by three different individuals to improve text fragments and correct possible errors, ensuring coherence.

An anonymous code was assigned to each participant (e.g., Informant one, two, three…) for confidentiality. During the pre-analysis phase, a thorough and repeated reading of the texts was performed, generating a list of ideas and an initial analysis plan. The analysis phase involved describing all useful data up to their interpretation, creating quotes, coding, and developing thematic categories to capture main themes. Text segmentation was carried out, analyzing categories in detail in terms of characteristics, dimensions, properties, and trends.

**Reliability and Validity Criteria:** Following Calderón’s criteria for qualitative research rigor and quality [16], this research demonstrates methodological and theoretical-epistemological adequacy. The chosen qualitative paradigm and the phenomenological perspective are suitable for understanding the experiences of nursing professionals in penitentiary institutions. The theoretical perspective is based on Callista Roy’s adaptation model, focusing on nursing’s goal to help patients adapt to their environment and promote health and well-being.

**Ethical Considerations:** Ethical approval was obtained from the Committee on Ethics in Human Research (CEISH) at the Universitat Jaume I de Castellón, under file number “CEISH/64/2023”. The project adhered to current regulations on data protection, including the European Regulation 2016/679 and the Organic Law 3/2018 on Personal Data Protection and the guarantee of digital rights.

This research complies with ethical standards, ensuring participant confidentiality, providing informed consent, and following all ethical norms governing research involving human subjects.

## 3. Results

Following the analysis of various nursing discourses regarding their perception of the HIV pandemic in penitentiary centers, four main themes were identified, as shown in Table 3.

The first theme corresponds to the view presented by the media on HIV and prisons, as perceived by nursing professionals. Generally, there is a considerable consensus that newspapers did not reflect the situation in prisons. There was a sense of disinterest on the part of the press towards prisoners, as they were marginalized individuals who did not matter to anyone. From the beginning, they were stigmatized with terms such as “homosexuals” and “drug addicts”.


*Informant 2: “…inmates were not seen as people who, even though the law protects them and all that, but no, they didn’t matter much because, in the end, they were drug addicts. So, especially in the beginning, it was only considered that the level was drug addicts and homosexuals, and then, well, they were marginalized people, so that mattered little”.*

*[F-61]*


The information in the press did not reflect the reality inside the prisons at all. It was often presented as highly biased and sometimes from a lack of understanding of what was actually happening, perhaps because there was some interest in not making it known.


*Informant 6: “…everything was very, very covered up, no, then over the years, it progressed, and more news came out, but well, at first, very few because it wasn’t in anyone’s interest that it came out, that there was HIV”.*

*[F-69]*


They aimed to portray a global, negative, and alarmist view, given the pandemic’s nature, comparing it to other pandemics like the Black Death, the most devastating in human history, labeling the HIV pandemic as the “plague of the twentieth century”.

The few news items that appeared were often associated with protests by inmates and even prison officials, demanding better hygiene and health conditions. Frequently, these news articles served as accurate reflections of the prevailing conditions within prison facilities, delineating issues such as overpopulation and substandard hygiene conditions.

The second theme focused on the perception of nursing work in prison centers, highlighting the pandemic’s impact on necessary changes, human, material, and organizational resource needs, as well as the acquisition of specific skills by nursing to address care.


*Informant 7: “…communication skills, skills in understanding different treatments, also because it depended a lot on, you know. Well, the experience we have, we have treatments that were not combined in the same type of dosage, but went with two or three types of treatments, so maybe one in the morning or another at night, fasting. So compliance was very complicated, but you have to have these clinical and scientific knowledge at the pharmacotherapeutic level so that you could inform the patient as much as possible, it’s also true”. *

*[F-47]*


Within this second theme, another emerging category pertained to the resources nurses had to contend with in carrying out their duties within the prisons, among which training, especially at the beginning of the pandemic, stood out. Quality training was provided through the Carlos III Health Institute, one of the leading centers in understanding this infection in Spain. Less experienced professionals acknowledge that there has been training but consider it limited. The National Plan on AIDS in our country also showed interest in training on HIV.


*Informant 3: “…I remember that exactly, I think it was in 1987, the first HIV courses given at Carlos III in Madrid, for prison personnel. Health professionals in prisons were the first to receive information, and then it has been updated, and in that sense, we can’t complain either”.*

*[M-61]*


Various programs established for monitoring and control also contributed. The treatments undoubtedly contributed to better control of this infection.


*Informant 5: “…the most important thing has been the treatments, they have enriched the laboratories, but thanks to the treatment, people still live”.*

*[F-63]*


The third emerging category of this second theme consisted of the main difficulties that professionals encountered within the environment. Prison institutions are primarily meant to guard prisoners and do not have a health-oriented purpose like a health center. Fundamental problems included access to inmates due to schedules and an attempt to violate confidentiality by wanting to know who was HIV-positive or not by surveillance officers.


*Informant 1: “…the regime, we all know, of course, we all know that in a prison, security takes precedence over anything else and then also over health”.*

*[M-64]*


The third theme revolved around the view of the patients attended to by these professionals, with two emerging categories: gender differences and inmates.

Generally, professionals state that there have been no differences in healthcare treatment between women and men regarding HIV infection in prisons. In other words, they operate on the concept of equality. Biologically identifiable situations, such as pregnancy, where women, unlike men, had the possibility of transmitting the infection to their offspring, were acknowledged.


*Informant 9: “…No differences, absolutely not. I think that in this sense, we are talking about a population that are addicts, and in that sense, they are all the same. There is no gender difference, that is, it is a drug addict, whether male or female”.*

*[M-55]*


Risk practices significantly influenced assistance and prevention, especially in terms of sexual relationships. Information and emphasis on the need for contraceptive methods were provided, along with the facilitation of all kinds of resources. In the case of pregnancies in positive mothers, the possibility of abortion began to be considered, not without controversy.

The second emerging category of this theme refers to the different attitudes of inmates who were initially reluctant to the possibility of being treated. At the start of treatments like Zidovudine (Retrovir^®^), there was high mortality despite having that single drug. In this early period of the pandemic, drug use (mainly heroin) through parenteral routes was the fundamental risk practice for acquiring the infection, and addiction conditioned an eagerness to consume in inmates that constituted their major concern, pushing everything else into the background.


*Informant 2: “…they rejected the treatment because, as it was given so late, it was given when they were dying, so the medications had a very bad reputation. Give me this, and I’m going to die”.*

*[F-61]*


The last theme raised was the nurses’ view of their professional role, personal experiences, and those of their surroundings during the HIV pandemic in prisons, with three emerging categories: the professional role itself, family and social relationships, and experiences lived.

The professional role was described as initially associated with information collection, as serological data before 1989 were not available, leading to extensive work through epidemiological surveys and blood extractions for serologies. The care role was also highlighted, emphasizing prevention, follow-up with a particular focus on treatment adherence, and providing support and comfort, constituting holistic care.


*Informant 4: “…direct contact with patients over long periods of time, it is nursing that dispenses that comprehensive and holistic care that other health professions do not delve into”.*

*[M-57]*



*Informant 10: “…you go with them accompanying them in the process, and feeling accompanied also helped them not to abandon the treatment”.*

*[F-47]*


The family and surroundings experience of nursing professionals initially involved a lot of fear of the possibility of contagion. The most experienced ones refer to this since, at that time, getting infected could mean signing a death sentence. This fear, over time, was seen to be mainly due to ignorance; it was a fear of the unknown. In other family relationships, discretion was exercised to avoid the suffering of relatives, revealing only what was necessary or nothing about these types of patients they worked with daily.Among the main negative experiences that can be linked to the beginning of the pandemic, we can mention fear, helplessness, sadness, loneliness, impact, or concern about having suffered a biological accident.


*Informant 1: “…I have also suffered a lot from the helplessness of seeing how very young people were dying, and you could practically not give them proper treatment beyond symptomatic treatment and acting a bit as a nurse, psychiatrist, or psychologist and listening to them and giving them a bit of comfort”.*

*[M-64]*


Among the positive experiences, which are linked to more recent times as the progression of the pandemic has been seen, professionals explain their own experiences highlighting hope, the learning it has represented, the change that has occurred, their own professional development, and, of course, the rewarding experience they have had the opportunity to enjoy.

## 4. Discussion

This research is undertaken with the intention of documenting a period in the history of prison nursing that witnessed significant adaptations to confront one of the most crucial pandemics we have encountered. Understanding, through the nursing experience, how these changes occurred and how resources and nursing practices adapted can be valuable for addressing future situations.

The HIV/AIDS pandemic in prison institutions has received limited coverage in the press, leading to a lack of reflection on the HIV phenomenon in prisons.

According to nursing professionals, the impact of HIV on incarcerated individuals brought about significant changes in healthcare, including human, material, and organizational resources, to implement an integrated adaptation process aimed at achieving an optimal state of health. These changes can be considered focal stimuli, according to Callista Roy’s theory for facing adaptation [16].

It also involved coping processes acquired by nursing professionals in terms of developing specific skills, abilities, and knowledge. Communication skills were highlighted to influence the need for proper adherence to treatments. The acquisition of pharmacotherapeutic knowledge to understand and teach about the adverse effects of many available treatments was emphasized. Nursing professionals’ creativity played a role in facilitating treatment adherence. The training and the establishment of preventive programs and treatments, which varied over time, were implemented for the adequate adaptation of patients.

Several inconveniences or difficulties arose that could lead to compromised coping, such as the established regimen in prison centers for security reasons, limiting nursing interventions in terms of schedules.

The HIV infection was evaluated, and informants generally agreed that no gender differences had been established from a professional perspective, emphasizing the concept of equality in professional treatment.

They did highlight that being a woman posed challenges regarding pregnancies in the case of HIV infection and the possibility of vertical transmission to the fetus, requiring information provision and preventive measures.

In general, incarcerated patients initially expressed reluctance to undergo treatments. Initially, this reluctance was due to the high mortality rates, and later, new drugs caused many adverse effects (focal stimuli according to C. Roy).

Among the main limitations of this research, it should be noted that convenience sampling burdens the methodology. Given the qualitative design, generalizations cannot be made, and the quality of an online-conducted investigation largely depends on the individual skills of the researcher.

This is where nursing intervention on adherence and the importance of following treatments properly influenced the adaptive process of patients to undergo appropriate treatment [16].

Informants described their professional role, personal experiences, family experiences, experiences in the environment, and their overall experiences regarding nursing practice in prisons, confirming self-concept or group identity and the social role in interacting with the environment, as described in Callista Roy’s adaptation model.

As future lines of research, it would be interesting to explore the experiences of other key figures, such as inmates who have undergone treatment in prison centers and surveillance officers, enriching the information content.

## 5. Conclusions

This research on the experience of Spanish prison nurses with the HIV pandemic between 1985 and 2023 reveals a period of profound changes and adaptations in healthcare within prison environments. Through the analysis of nursing discourses, four key themes were identified that addressed the perception of the press, nursing work, the vision of patients, and the personal experience of professionals.

The first theme highlights the discrepancy between the reality experienced in prisons and the media representation of HIV. The press, in many cases, did not reflect the internal situation, showing disinterest and stigmatization towards inmates. This lack of coverage affected public awareness of the pandemic in prison settings and contributed to the marginalization of inmates.

The second theme emphasizes the evolution of the nursing role and the challenges that emerged during the pandemic. Training, the acquisition of communicative skills, and the management of antiretroviral treatments were crucial elements. However, limitations of the prison environment, such as restricted schedules and a lack of focus on health, presented significant challenges.

The third theme addresses the nursing view of patients, highlighting equality in treatment between men and women but recognizing biological differences, such as the risk of vertical transmission in pregnant women. Initial resistance attitudes from inmates toward treatments, influenced by fear and distrust, are also highlighted.

The last theme reveals the personal and professional experiences of nursing, showing a range of emotions from fear and helplessness to hope and learning. The adaptation of nursing professionals over the decades reflects resilience and coping ability in challenging environments.

In conclusion, this research highlights the valuable contribution of prison nursing in adapting to and coping with the HIV pandemic. Lessons learned and shared experiences provide a solid foundation for facing similar challenges in the future and underscore the importance of quality care in prison settings.

## Figures and Tables

**Table 1 healthcare-12-00184-t001:** List of guiding questions for the discussion groups.

How would you describe the way the situation in penitentiary institutions regarding the HIV/AIDS pandemic has been portrayed through the Spanish print media?
What would you highlight as the most significant aspect regarding the impact of HIV in penitentiary centers?
What role has nursing played in the HIV pandemic within penitentiary institutions?
How was your experience working for the first time with a patient infected with HIV or even suffering from AIDS? (in different spheres: professionally, personally, in social relationships, etc.).
Do you believe that there has been any gender-related difference between male and female inmates regarding HIV in prisons?
What resources have you had available for providing nursing care to HIV/AIDS patients?
What difficulties or obstacles may have influenced your work regarding HIV-infected individuals in the penitentiary environment?
How would you summarize your own experience regarding the HIV pandemic in penitentiary institutions?

**Table 2 healthcare-12-00184-t002:** Characteristics of participants in the discussion groups.

		Group 1	Group 2
Age (in years)	Mean	62.5	48.2
	Minimum	57	44
	Maximum	69	55
Seniority in the profession (in years)	Mean	39.7	27.2
	Minimum	36	23
	Maximum	48	30
Seniority as prison nurses (in years)	Mean	35.5	24.5
	Minimum	33	17
	Maximum	38	27
Marital status	Single	1	1
	Married	4	2
	Separated	1	2
	Widowed	0	1
Employment status	Active	2	6
	Retirement	4	0

**Table 3 healthcare-12-00184-t003:** Data analysis structure.

*Codes*	*Categories*	*Themes*
Alarmistic; ignorance; disinterested; interested; protests; little impact; biased; taboo.	News	Press perception of HIV in prison.
Changes; human resources; material resources; organizational resources; skills.	Importance of HIV	Vision on nursing work in prisons
Training; programs; treatment.	Media	
Environment; people.	Difficulties	
Attitudes; demands.	Inmates	View of the patients they care for.
Contraception; pregnancy; equality; risk practices.	Gender	
Adherence; support; care; information; prevention; follow-up.	Nursing role	Nursing’s own vision
Discretion; fear; caution; professionalism.	Family and environmental experiences	
Positive; negative.	Experiences	

## Data Availability

The data presented in this study are available upon request to the corresponding author.

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
