# Peer review of "The Experience of Spanish Prison Nurses with the HIV Pandemic"

_healthcare, 2024, doi:10.3390/healthcare12020184_

Round 1

Reviewer 1 Report

Comments and Suggestions for Authors

Dear authors!

Thank you for the opportunity to read your work. The paper addresses the changes and adaptations in Spanish prisons due to the HIV/AIDS pandemic. It seeks to understand the experiences of nurses in this context. I reviewed this manuscript with great interest, but there were many problems for publication. Generally, the paper has many shortcomings in elements from the title to the conclusion. The title should be better conceived; it even has irregularities, for example, writing 'aids' in lowercase and using inappropriate expressions such as 'nurses point of view'.

The abstract is entirely poorly written. It is not clear why the research is being conducted, nor is the purpose stated. The main results are not clearly presented; the abstract lacks an explanation of why we care about the results and what new information we have learned.

The introduction starts inappropriately with some purpose and a rhetorical question. The content is based on 11 references, of which 6 are used in just one paragraph. It is unclear: Why is the research problem important? How does your research build upon previous studies? What are the primary (and if any, secondary) hypotheses, and do they stem from a theory/model? What are the theoretical and practical implications of the research?

The convenience sampling burdens the methodology, which you have not mentioned in the limitations. The results section is too extensive, containing irrelevant data, especially in Table 2, where it's unclear what to conclude if one group has 4 retirees while another has none, other than that the sample is uneven. The discussion is overly brief. It is problematic that the limitations of the research, which could question the results and their interpretation, are not mentioned. There is no conclusion. The literature is cited without adhering to the guidelines. Unfortunately, I believe that in its current form, the paper is not sufficiently robust for publication.

Comments on the Quality of English Language

It is necessary for a native speaker to read the paper.

Author Response

Thank you for the opportunity to review our manuscript, and for the contributions you have made, which we have considered and modified

Next, I respond to your contributions and discuss the changes considered after your review

  • The title is modified to: 'The experience of Spanish prison nurses during the HIV pandemic.'

  • The abstract is completely revised.

  • The introduction begins inappropriately with a specific purpose and a rhetorical question. The question is modified to enhance clarity regarding the significance of the topic, both from a historical perspective of Spanish prison nursing and its relevance for future pandemics."

  • The content is based on 11 references, with 6 of them being used in a single paragraph. It is clarified at the end of the introduction that due to the lack of scientific research, historical information sources such as the Spanish print media have been utilized to reflect on the referenced information regarding the pandemic in Spanish prisons.

  • It is not clear: Why is the research problem important? It is significant from a historical perspective for Spanish prison nursing and from a public health standpoint in addressing future pandemics (as reflected in the introduction).

  • How is your research based on previous studies? No specific studies addressing this topic have been found, as described in the introduction.

  • What are the primary (and, if any, secondary) hypotheses?  No hypotheses are proposed in this research;  Its sole objective is to reflect the experiences of Spanish prison nurses in the face of the HIV/AIDS pandemic.

  • Do they derive from a theory /model?  What are the theoretical and practical implications of research?  The implications are reflected in the conclusions added at the end of the discussion.  There is a theoretical model that resembles the conclusions, namely callista Roy's adaptation model, to which reference is made."

  • Convenience sampling overloads the methodology, which has not been mentioned in the limitations (included as a limitation).

  • "The results section is too extensive and contains irrelevant data, particularly in table 2, where it is unclear what to conclude if one group has four retirees while the other has none, other than the unequal sample size.  The sample consists of two groups: one made up of professionals who faced the pandemic at its onset when there were no treatments available (hence some are already retired), and the other made up of professionals who dealt with the pandemic when antiretroviral drugs were available.  They therefore constitute homogeneous groups within themselves and, at the same time, heterogeneous among themselves (as indicated in the section on methodology).

  • The discussion is too brief.  It is problematic that the limitations of the research are not mentioned, which could call into question the results and their interpretation.  Limitations are added.  There is no conclusion.  Conclusions are added."

"Once again, thank you very much for this opportunity.  Awaiting your response, receive a cordial greeting."

Reviewer 2 Report

Comments and Suggestions for Authors

Thank you very much for submitting your manuscript to MDPI.  This is an interesting study and will be valuable to the readers.  It is not clear whether there were any limitations in this study. Was it difficult to recruit participants, was there any form of incentive given to the 'informants' to participate in the study? It will be useful if some possible limitations are included in the discussion section of the manuscript.

Author Response

Gracias por la oportunidad de revisar nuestro manuscrito y por las contribuciones que ha realizado, que hemos considerado y modificado.

"A continuación, respondo a tus aportes y analizo los cambios considerados luego de tu revisión.

"No está claro si hubo limitaciones en este estudio. Se incluyen las limitaciones del estudio. ¿Fue difícil reclutar participantes? ¿Se dio algún tipo de incentivo a los “informantes”? para participar en el estudio? Se observa en el proceso de selección que no hubo dificultad en el reclutamiento, y no se incentivó la participación de los informantes. Será útil incluir algunas limitaciones potenciales en la sección de discusión del manuscrito. Las limitaciones están incluidas."

Una vez más, muchas gracias por esta oportunidad. Esperamos su respuesta, reciba un cordial saludo.

Reviewer 3 Report

Comments and Suggestions for Authors

1.     The abstract does not highlight the specifics of your research or findings. The author should rewrite the abstract to address the problem, Aim, Methods, Results, and Conclusion.

2.     The introduction section should be elaborative and explain the background and need of the proposed work in greater detail along with its necessity.

3.     The primary objective of this research is to describe the experiences of nursing personnel in Spanish prison institutions regarding the HIV/AIDS pandemic, how will it help in bridging the existing gap.

4.     Since this work presents a particular group’s POV on the given situation, how can it be generalized and accepted?

5.     What led the author to come up with this kind of research initiative and how will it help the upcoming times?

6.     The author should work on the result section as the results presented are not self-explanatory and do not highlight the proposed work's impact.

7.     The author should check for spelling and grammatical mistakes in the paper.

Comments on the Quality of English Language

Moderate editing of English language required

Author Response

Thank you for the opportunity to review our manuscript, and for the contributions you have made, which we have considered and modified.

Next, I respond to your contributions and discuss the changes considered after your review.

  1. "The abstract does not highlight the details of your research or findings. The author should rewrite the abstract to address the problem, objective, methods, results, and conclusion."

    Modified

  2. "The introduction section should be elaborated to explain the background and need for the proposed work in more detail, along with its significance. At the beginning of the introduction, the need and importance of reflecting the history of Spanish prison nursing, as well as the usefulness of coping with a pandemic to act in future similar situations, are clarified. The background is reflected in the introduction."

  3. "The main objective of this research is to describe the experiences of nursing staff in Spanish prisons during the HIV/AIDS pandemic and how it will help bridge the existing gap. The objective is reformulated based on the reviewer's suggestion, and the authors' intentions to bring visibility to an initially overwhelming situation in an unfamiliar environment for society in general are clarified."

  4. "Since this work presents the viewpoint of a particular group on a given situation, how can it be generalized and accepted? The difficulty of generalizing results from the qualitative design is discussed in the limitations."

  5. "What led the author to conceive this type of research initiative and how will it help in the times to come? At the beginning of the discussion, the intention to contribute to documenting a period in the history of prison nursing that brought about numerous changes to adapt and cope with one of the most significant pandemics we have faced is presented. Understanding from the nursing experience how all these changes have occurred and how they have adapted can be useful for facing future situations."

  6. "The author needs to work on the results section as the presented results do not explain themselves and do not highlight the impact of the proposed work. The results have been reviewed and revised, although we believe they emphasize the importance experienced during the pandemic, the objective of the work."

  7. "The author should check for spelling and grammatical errors in the article. The spelling and grammatical errors of the article have been reviewed.

  Once again, thank you very much for this opportunity. We look forward to your response, please receive a cordial greeting.

Round 2

Reviewer 1 Report

Comments and Suggestions for Authors

Dear authors, I thank you for the effort and additional corrections. The work has been significantly improved. Please check the reference list and make further corrections.